# Voluntary sports programs for individuals with mental health disorders: The trainer's view

Florence Epiney[1,2], Frank Wieber[3], Daniela Loosli[1], Hansjörg Znoj[2,4], Nikolai Kiselev[5,6]*

1 PluSport Bern Gruppen, Bern, Switzerland, 2 Department of Psychology, University of Bern, Bern, Switzerland, 3 School of Health Professions, Institute of Health Science, Zurich University of Applied Sciences ZHAW, Winterthur, Switzerland, 4 Department of Psychology, University of Konstanz, Konstanz, Germany, 5 Swiss Research Institute for Public Health and Addiction (ISGF), University of Zürich, Zurich, Switzerland, 6 PluSport, Umbrella Organization of Swiss Disabled Sports, Volketswil, Switzerland

* kiselev@plusport.ch

**Data Availability Statement:** According to the informed consent, we promised to the participants (trainers) that the interview transcripts will be storage according the data protection requirements by PluSport and will not be published/uploaded

## Abstract

There is strong evidence that physical activities (PAs) are an important factor in increasing and maintaining mental health as well as in preventing relapse after mental health disorders. Physical activity is an important part of the treatment program in psychiatric hospitals. However, when individuals with mental health disorders (IMHD) leave the hospitals in Switzerland (CH), there are few possibilities to do physical activity in a given setting. One of them are voluntary sports groups for individuals with mental health disorders (SGPSY), which have been growing continuously in CH since 2016. Yet, little is known about these groups and their training settings. Therefore, the present study explores challenges, barriers, and enablers for participation in SGPSY from the point of view of the trainers of these groups. Additionally, as the sustainable implementation of SGPSY relies on the trainer, the study aims to identify reasons/motivations as well as the personality characteristics of the SGPSY trainers. Semi-structured interviews were conducted with 15 trainers of SGPSY in CH during spring 2022. Interviews were audiotaped, transcribed, and analyzed using thematic analysis in nVivo. Participants identified several intrapersonal (lack of motivation and fitness, mood problems, etc.), interpersonal (conflicts between participants), and structural barriers (time/ location) that hinder IMHD from participating in SGPSY. The participating trainer reported that trainer might be helpful in overcoming the barriers by supporting IMHD as enablers. They rate social skills to be essential for the successful management and organization of SGPSY, as well as the ability to set boundaries to protect one's private life and sports skills expertise. The reasons for their engagement as trainers of SGPSY were the satisfaction of doing sports with IMHD and to improve the physical activities habits of IMHD. The findings of the study highlight the need to upskill the trainers of SGPSY in order to improve recruitment of the future trainers of SGPSY by focusing on the assessment of appropriate personality characteristics of trainers and their motives. Additionally, these findings should be integrated in the educational materials of Swiss disabled sports systems. Further research should validate the results from SGPSY participants' point of view.

anywhere to the open access. This is important due to the fact that trainers are working with a very sensitive group – individuals with psychiatric disorders. A single statement might be interpreted in the wrong way and influence the group dynamics etc. To ensure the trainers speak freely, it was necessary to accept this obligation. And for these reasons, we are strongly reluctant to publish the transcripts in any open access data banks. However, we are happy to share the transcripts (in German with editor and/or reviewers) under condition that they will not publish it as open access. In addition to contacting the corresponding author, the following organizations can be requested for data access: upon reasonable request at PluSport, Umbrella Organization of Swiss Disabled Sports, Volketswil, Switzerland [mailbox@plusport.ch] or at the Department of Psychology, University of Bern, Bern, Switzerland [info@psy.unibe.ch].

**Funding:** This study was supported by a grant from the Biäsch-Stiftung (Ref. 2021-13 / "Sport für Menschen mit psychischen Beeinträchtigungen in der Schweiz» / by NK) [https://biaesch-stiftung.ch/ ]. The funding institutions took no part in the design of the study and collection, analysis, and interpretation of data or in writing the manuscript. The opinions expressed and arguments employed herein do not necessarily reflect the sponsor's views.

**Competing interests:** The authors have declared that no competing interests exist

# 1. Introduction

There is strong evidence for the positive influence of physical activities (PA) on physical and mental health [1–5]. Thereby, PA not only reduces the risks of mental health disorders but might be applied as an effective instrument for treating mental health disorders and preventing relapse [1, 4, 6]. Several studies showed that PA reduces symptoms of major depression [7–10], schizophrenia [11, 12], bipolar-disorders [13] and anxiety as well as stress-related disorder [14]. In fact, as Stubbs et al. [15] observed in their meta-review, physical activity effectively supported the treatment of individuals with severe mental health disorders, sometimes as much as medication in individuals with schizophrenia and major depressive disorder.

Given that PA reduces the symptoms of MHD and improves the general mental state of IMHD and knowing that IMHD shows lower lever of PA [16, 17], it is evident that PA activities should be promoted among this population. As a result, almost every psychiatric hospital in Switzerland implemented PA in their treatment programs [18].

However, the situation is completely different outside the hospitals. IMHD have no restrictions regarding voluntary participation in sports or PA, but particularly individuals with severe MHD massively reduce their PA in everyday life or completely stop it [17, 19]. Besides the well-known motivational problems (that correlate with the symptoms of MHD), there are several other MHD-specific reasons (e.g., fear of stigma, low level of physical fitness, side effects of medication, infrequent participation, low ambiguity tolerance, previous negative experience with sports [e.g., mobbing at school or showing unsatisfactory sports results in school]) for the low level of PA in IMHD [5, 12, 20–24]. At the same time, it is known that some aspects (e.g., social support, social contacts with other participants, shaping of social identity, or the experience of competency) can improve the level of PA in IMHD [25–30]. Hence, specific PA programs that are adjusted to the needs and abilities of IMHD represent a promising measure to boost their participation in PA.

Yet, just less than a decade ago, there were no adapted PA programs for IMHD outside the hospitals in Switzerland [22, 24]. In response to the lack of adapted PA for IMHD, PluSport, the Umbrella Organization of Swiss Disabled Sports, started the development of specific sports groups for this target population and active integration of IMHD into the existing structures of Swiss Disabled Sports in 2015. Besides adapted sports camps, there are already over 20 specific sports groups of IMHD that have been offered by PluSport by the end of summer 2022 [31]. The number of active participants with MHD increased from less than 250 to almost 600 between 2015 and 2019 within the organization [32]. Despite the importance of sports for IMHD, the number of IMHD in the Swiss population [33], and the growth trends within Swiss Disabled Sports, it might be reasonable to assume that the number of sports groups will increase in the future.

Given the growing number of sports groups and camps for IMHD in Switzerland, the number of trainers for these programs also need to increase. To ensure the quality and security of the sports programs, trainers of the sports programs in Switzerland are required to gain and prolong a specific sports education offered or supervised by the Swiss Federal Office of Sport (SFOS). The SFOS is a service, education, and training center for elite, professional, and amateur sports. As an education center, it develops knowledge and imparts the skills and proficiencies required to practice and teach professional and amateur sports. The training and advanced sports educational programs for the trainer of sports clubs and sports groups are provided or supervised by SFOS in the framework of J+S and esa programs (J+S = Youth + Sports promotion program; esa = adult sports). The trainers for sports programs for individuals with disabilities are educated within the framework J+S or/and esa-programs as well.

However, these educational programs are strongly oriented toward regular sports (for individuals without disabilities) or the « classical » well-known physical or intellectual disabilities. Based on the fact that the voluntary sports programs for IMHD are a relatively new phenomena, there is almost no literature about the organizing and training of this population. Except one: all regular educational materials for sports trainers mention the IMHD very marginally or do not mention it at all [34–38]. Although other specialist literature regarding movement and exercise therapy exists, this literature is limited to expert literature for readers with medical and paramedical backgrounds and applies only to the treatment and/or inhouse setting [39–44]. At the same time, trainers of sports groups and their characteristics are crucial factors for the functionality of a successful sports group [38, 45–55].

In sum, trainers of sports groups are one of the major elements of well-functioning voluntary sports programs. However, there is almost no information and knowledge on how to train and organize voluntary sports groups for IMHD. Yet, this information and knowledge would be very desirable to address the special needs of the participants by the trainer of the voluntary sports groups for IMHD. Additionally, it might improve the trainer's education to prepare them for the work with IMHD providing the quality and security of the voluntary sports groups for this population.

### 1.1. Aims

The present study aims to explore and analyze the SGPSY in Switzerland from the trainer's point of view. Therefore, five guiding questions were defined:

1. Are and if yes what are the challenges and barriers that inhibit/hinder the participation of IMHD in SGPSY?

2. Are and if yes what are the facilitators for the participation of IMHD in SGPSY?

3. What are the personal characteristics of trainers of SGPSY?

4. Are and if yes what are the reasons/motives for working as a trainer of SGPSY?

5. Are and if yes what are the outcomes (at a personal level) for trainers of SGPSY?

## 2. Materials and methods

The study aimed to explore the perspective that trainers of the voluntary sports groups for IMHD have on their work and situation in these groups. Based on the fact that there were no comparable studies or other comparable publications analyzing the point of view of the trainers from voluntary sports groups for IMHD, a cross-sectional, qualitative design was applied using in-depth semi-structured interviews.

### 2.1. Setting

In Switzerland, physical activities and sports programs for individuals with disabilities are provided by three nationwide organizations: PluSport–the umbrella organization for disabled sports, Procap–the association for disabled persons (offers sports only partly), and Wheelchair sports (Rollstuhlsport). The range of physical activities and sports programs is wide. Almost 700 sports groups (regular programs (e.g., weekly) and over 150 sports camps (usually 1–2 weeks lengths)) cover nearly every possible type of sport [32, 56]. In each sports group or camp, at least the head trainer must have a valid qualification as a "Coach for disability sports" [Behindertesportleiter] (depending on the number of participants and type of disabilities:

more participants or/and more severe disabilities = more qualified coaches are required). All assistant trainers must complete the qualification as an "Assistant for disability sports" [Assistenzmodul für Behindertensport]. For these reasons, almost every person providing voluntary physical activities and sports for individuals with disabilities is registered in PluSport, Procap or Wheelchair sports data bank allowing to approach them directly.

## 2.2. Participants

Following the previous research in the field and in accordance with expert literature, we aimed to interview 15 trainers that have been working for at least one year as a trainer of a regular voluntary sports group for IMHD using maximal variation sampling [57–68]. Given that there were only 17 main trainers (incl. former) with at least one-year of training experience in a regular voluntary sports group for IMHD, at the moment of interviewing, we aimed to reach at least 10 of them. In case of recruitment difficulties with the main trainers, we aimed to compensate up to 5 main trainers with assistant trainers who have at least two years of experience in a regular voluntary sports group for IMHD.

## 2.3. Interviewer

All interviews were conducted in (Swiss-)German by a psychology master's student from the University of Bern (in the last study year). The interviewer has been involved in the project since the grant application. She profoundly understands the setting because of her long-term involvement as a qualified and educated trainer (Behindertensportleiter = disability sports trainer) in various sports groups and camps for individuals with disabilities. Moreover, she had prior interviewing experience in other research projects and was additionally trained in practicing open-ended interviewing with the target population of the current study.

## 2.4. Recruiting

Given that PluSport is the only organization that offers voluntary sports programs for IMHD in Switzerland [69], these groups' trainers and assistant trainers were identified in the data bank of PluSport. After the exclusion of the absent persons and those who were not active during the period impacted by the coronavirus outbreak and related restrictions, 21 persons were identified. On the starting day, March 30th, 2021, they received a personalized email with a brief project description and an invitation to participate in the interview. Moreover, a reminder was sent one month later, on the 30th of April 2021.

## 2.5. Process

Interviews were conducted in (Swiss-) German. All interviews were audiotaped and transcribed verbatim. Informed consent was obtained orally (since we did not ask about any personal or health-related questions, there was no necessity for written informed consent). Before focusing on the study's topic, the participants' social-demographic information was assessed.

The interviews took place between the 6th of April and the 12th of June 2021. Because of the pandemic, all interviews were conducted via Zoom.

There were no monetary compensation or other incentives for participation.

## 2.6. Analysis

We used applied thematic analysis as described by Guest, MacQueen, and Namey [57]. Initially, the research team examined the dataset to identify central themes in the data. Thereafter, a student in the last year of the study program for the master's degree in psychology created

the codes and integrated their groupings into the broader concept under the supervision of a senior researcher. The research team reviewed and mutually agreed on the coding framework as the next step. Thereafter, the final coding was employed and implemented throughout the data. The ambiguous or uncertain cases were discussed with a senior researcher. Finally, a person from the research team, who was not involved in the first step (to eliminate possible bias), coded 20% ($n = 3$) of the dataset according to the final coding framework. The interrater reliability was k = .90, which can be interpreted as an excellent agreement [70].

The coding was performed using NVivo 12 software (Version 12, QSR International). It is important to note that it was possible to allocate one statement to several codes since each statement could contain information relating to several themes.

### 2.7. Ethics

Ethical approval was not required since the present study does not fall within the scope of the Swiss Human Research Act. However, the project has been approved by the ethics committee of the University of Zurich.

## 3. Results

### 3.1. Demographics

Interviews with the 15 participating adult trainer ($N = 15$) lasted between 32 and 50 min. The characteristics of the participants are presented in Table 1. The trainer were allocated to different PA as follows: five ($n = 5$) were trainers of polysports groups, four ($n = 4$) from climbing groups, four ($n = 4$) from specific ball sports groups, one ($n = 1$) from ping-pong, and one ($n = 1$) from the nordic walking group.

### 3.2. Results related to guiding questions

To support a general overview of the results, we report the findings along the five guiding questions (s. chapter 1.1. "Aims"), After a short summary at the beginning of each subsection, the results are presented in detail.

**3.2.1. Challenges and barriers.** Based on the data, three main categories of barriers and challenges were identified and summed up under three categories: intrapersonal, interpersonal, and structural (barriers and challenges). In sum, intrapersonal challenges and barriers were mentioned by each participant ($N = 15$) as a problem for IMHD for participating in voluntary sports groups for IMHD. Every second participant reported interpersonal and/or structural challenges and barriers as a problem for IMHD for participating in SGPSY ($n = 9$ and $n = 8$, respectively). Details are presented in Table 2.

**Table 1. Demographics of the interviewees.**

| Characteristics | Total Sample ($N = 15$) | Trainer ($n = 12$) | Assistant Trainer ($n = 3$) |
|---|---|---|---|
| Age, median *(range)* | 36 *(25–67)* | 34 *(25–67)* | 40 *(36–59)* |
| Gender (%) | | | |
| Female | 8 (53%) | 6 (40%) | 1 (7%) |
| Male | 7 (47%) | 6 (40%) | 2 (13%) |
| Professional background (%) | | | |
| Secondary education | 3 (20%) | 2 (13%) | 1 (7%) |
| Tertiary education (completed) | 9 (60%) | 7 (47%) | 2 (13%) |
| Currently in tertiary education (Psychology) | 3 (20%) | 3 (20%) | 0 (0%) |
| Number of years in function, median *(range)* | 4 *(1–27)* | 3.5 *(1–27)* | 10 *(3–12)* |

**Table 2. Frequencies and percentages of treatment barriers.**

| Challanges and barriers | Frequency | Subcategory | Frequency |
|---|---|---|---|
| Intrapersonal | 15 (100%) | | |
| | | *Mental state* | 11 (73%) |
| | | *Motivation, listlessness* | 11 (73%) |
| | | *Physical* | 8 (53%) |
| | | *Medication* | 5 (33%) |
| Structural | 9 (60%) | | |
| | | *Time + place* | 6 (40%) |
| Interpersonal | 8 (53%) | | |
| | | *Group composition* | 8 (53%) |

*Note*. Full sample *N* = 15; Challenges and barriers mentioned less than five times are not included.

*3.2.1.1. Intrapersonel challanges and barriers.* Participants' (of the sports groups / IMHD) mental state was the interpersonal barrier that was mentioned most (*n* = 11) to participate in voluntary sports programs. The trainers explained that from their point of view, mood lability and other complex mental conditions make it challenging or impossible for participants to participate in sports programs and could even lead to discontinuing the participation in the sports group:

> "And the second aspect, which is very challenging for them, is probably also that they are then a bit, depending on the phase, also in their own world, and then there are a lot of these people who somehow bring their own baggage with them and are perhaps easily irritable and easily blow up." (ID_14)

[Quotes were translated from German for the present article]

Furthermore, in connection with mental state, the lack of motivation and listlessness were often (*n* = 11) highlighted as possible barriers to participation. Moreover, trainers (*n* = 8) named physical requirements as a challenge for the participation in the sports groups for IMHD, as physical fitness is often lower in the population of IMHD compared to the average population. A lack of fitness, which also could have resulted through previous negative experiences of the IMHD with PA (for instance, during school), makes, from the point of view of trainer, participation as well as the PA itself more challenging as the following section demonstrates:

> "Then, they often can't imagine that climbing per se is possible at all. Precisely perhaps if there is still some excess weight involved. Or quite the opposite, quite thin for example, anorexic for example. Where maybe the strength at the beginning is just not so present." (ID_06)

As an additional barrier, five trainers mentioned IMHD's medication (*n* = 5). From their point of view, it makes the active participation more difficult and challenging due to side effects (e.g., affecting the movement capacity, perception of movement, etc.).

*3.2.1.2. Structural challanges and barriers.* Regarding structural factors which make participation in the voluntary sports group more challenging, trainer frequently (*n* = 6) named the time and place of the sports program as a barrier:

"Well I think that the biggest challenge is for them to be able to balance that with their occupations, their job. The offer was from 4pm to 6pm. And most of them are in a supervised structure and then they get time off and then somehow. . . Often it is an organizational effort that it really works out every week and that did not work out for all of them. And that was then sometimes difficult or it needed a certain amount of time to get it started." (ID_03)

*3.2.1.3. Interpersonal challenges and barriers.* With respect to interpersonal factors, trainers mentioned that difficulties between persons hinder the participation of IMHD. More than half of the trainers ($n = 8$) said that the group composition is a possible hindering factor. They explained that group constellation prevents IMHD from taking part in the sports groups as follows:

"However, they are not uncomplicated people per se. That means that there is also friction and then someone does not come for two weeks because they have had a fight somewhere and so on." (ID_02)

*3.2.1.4. Barriers and challenges only rarely mentioned.* Besides the barriers and challenges only once mentioned (e.g., pandemics), several other barriers were rarely discussed. Three trainers stated stress ($n = 3$) as a possible reason for the absence of IMHD from the training. However, depending on the respondent, the stress description was related to different issues like "putting oneself under pressure" or "stress as a result of the participation in general". Some participants ($n = 3$) reported difficulties in participating due to the low cognitive resources or the way of thinking. These examples were mainly related to older persons that needed more time to understand what to do in training. Furthermore, four trainers ($n = 4$) brought up fears and worries (about one's own look [because of adipositas] or because someone did not do any sports for decades) as a challenge to participate. Finally, a lack of financial means ($n = 4$) to cover the costs for participation and problems with access to the sports programs ($n = 4$) by not knowing about the sports groups or by lack of them in the surroundings were mentioned as barriers and challenges in the interviews.

**3.2.2. Enablers.** The following subchapters present an overview of the factors that might support or boost participation in SGPSY. The identified enablers and their subcategories are shown in Table 3 and summed up below.

*3.2.2.1. Trainers of SGPSY.* All but one of the respondents ($n = 14$) mentioned that from their perspective, the trainer him/herself acted as a supporting factor for participation in a

**Table 3. Frequencies of enablers for IMHD to participate in SGPSY.**

| Enablers | Frequency | Subcategory | Frequency |
|---|---|---|---|
| Trainers | 14 (93%) | | |
| | | *Relationship building* | 9 (60%) |
| | | *Stability, constancy* | 7 (47%) |
| | | *Social support* | 9 (60%) |
| Group composition | 12 (80%) | | |
| | | *Team-, group cohesion* | 11 (73%) |
| External social support | 7 (47%) | | |
| | | *Through persons in the immediate environment* | 5 (33%) |
| Sports program | 7 (47%) | | |

*Note.* Full sample $N = 15$; Enablers mentioned less than five times are not included

sports program for IMHD. Among other things, relationship building was highlighted as a supportive factor, which was suggested by more than half of the interviewees ($n = 9$). They explained that from their point of view, being a kind of attachment figure as a trainer for the participants promotes participation in physical activity.

> "Yes, certainly social contacts, I think that's something very important. (. . .) And the coming together also. This group has been around for a long time, they have really grown together. And that certainly helps them. And I also think that our leadership suits them. It helps them to come along and to enjoy it. The interaction with each other." (ID_13)

In addition, the stability and constancy of the trainers (trainer team) were emphasized as an essential factor ($n = 7$), which, from the trainers' point of view, strengthened participants to take part in the SGPSY. The trainers explained that when the same person led the group and the group ran in a similar way, it supported participation by IMHD for as follows:

> "And above all, the trust that the group is relatively stable, that there is not a different person every time, but that the same faces appear again and again. This way, they know what they can expect. So that there is a kind of security or a safe framework for them." (ID_11)

According to the trainers, constancy and stability of the trainers' team help IMHD to build up and strengthen their trust in the trainers and thus increase the likelihood of participation. According to the trainers, this process was also brought about by developing social support, which was presented as a further reinforcing factor for participation by nine interviewees ($n = 9$).

*3.2.2.2. Group composition.* The group composition as an enhancing factor was mentioned 12 times ($n = 12$). All but one ($n = 11$) referred to group cohesion and mutual dependence as a booster for participation. Consequently, trainers frequently reported higher participation rates if the IMHD already knew someone who was also participating in SGPSY and if there was a comfortable interpersonal atmosphere within a sports group or if there were a harmonious relationship among the participants of a sports group:

> "I also think that it helps if you can create a supportive community, in our case. It is not just about doing sports together, but also about creating friendships or a sense of belonging, and that people then seek out the sports program because of that. And if you succeed in supporting this, then of course it also helps them to stick with it and come to training regularly." (ID_08)

*3.2.2.3. External social support.* Several trainers ($n = 7$) mentioned that for participants, having an external support coming from persons in their circle, was another reinforcing factor for participation in the SGPSY. If the persons in their immediate environment (e.g., friends, family, caregivers, etc.) empowered or facilitated the IMHD to participate in the SGPSY (e.g., by providing transport), encouraged or reminded them to participate, it increased the likelihood of participation.

*3.2.2.4. Sports program.* The sports program (contents) was described as a strengthening factor for participation by almost half of the interviewees ($n = 7$). In the case of an appropriate sports program where the difficulty level matched the participants' abilities and motivation, participants usually experienced sports and their own abilities positively. These positive experiences increase their self-efficacy and motivate participants to return to the sports program again:

"Also this encouragement and the "I have faith in you". Especially in the climbing area, that is something crucial, and I really have the feeling that they go home stronger than before. I notice that they were given confidence and that they were able to cope with it. That is also a high level of self-confidence that they need in order to climb to heights of 10, 12, 17 meters and it simply benefits them, yes. But I think the consistency helps them, plus the knowledge that "I'm fine after climbing." It is almost a little addictive. That is nice. I think that the sport contributes insanely to that, that is very obvious. More than ball games or anything else, so I say this now, this is not colored neutrally. There is simply a satisfaction in climbing, when I achieve something, already extremely high. To gain self-confidence like that." (ID_05)

*3.2.2.5. Enablers only rarely mentioned.* Besides enablers only once or twice mentioned (e.g., participation in sports events), proactive and individualized communication of the trainer by asking/inviting the participants to the next training session (e.g., using WhatsApp) was mentioned four times ($n = 4$) as an important factor to improve the participation rate of IMHD.

**3.2.3. Personal characteristics of trainers of SGPSY.** To assess the what personal characteristics SGPSY trainer should have, we used the competence model of Zeuner and Hummel [71] based on four domains (professional expertise, methodological expertise, self-expertise, and social skills). This model is widely applied in the Swiss sports education systems. Additionally, we tried to assess the "negative characteristics" that would not be suitable for a trainer of the SGPSY (negative competencies). The results are presented in Table 4. The single categories are described below.

*3.2.3.1. Professional expertise.* Professional expertise is defined as the existence of specific professional skills and knowledge needed to accomplish a particular task [71]. The majority of the interviewees ($n = 13$) said that they consider didactic skills to be the most important. They explained that specific knowledge about the trainer's tasks, and taking into account the different performance levels within a group and finding as well as defining suitable solutions for the entire group were essential:

**Table 4. Frequencies of the characteristics of the trainers.**

| Personal characteristics | Frequency | Subcategory | Frequency |
|---|---|---|---|
| Professional expertise | 14 (93%) | | |
| | | *Didactic skills* | 13 (87%) |
| | | *Sport* | 7 (47%) |
| | | *Knowledge about mental disorders* | 7 (47%) |
| Methodological expertise | 14 (93%) | | |
| | | *Transformational leadership* | 13 (87%) |
| Self expertise | 11 (73%) | | |
| | | *Enjoyment of sports & leadership* | 7 (47%) |
| | | *Humanistic view* | 7 (47%) |
| Social skills | 15 (100%) | | |
| | | *Personality traits* | 15 (100%) |
| | | *Social interactions with IMHD* | 11 (73%) |
| | | *Delimitation* | 8 (53%) |
| Negative characteristics | 14 (93%) | | |
| | | *"An end in itself"* | 5 (33%) |

*Note.* Full sample *N* = 15; Personal characteristics mentioned less than five times are not included

"She comes just for the walk and then we have to make sure that we don't walk too fast, I have to slow down the group again, because we have some that march off with determination and then someone falls off the back." (ID_07)

Almost 50% of the trainers (*n* = 7) emphasised knowledge about the sport itself as an important professional expertise, including knowledge about different exercises of a sport:

"Sure it's also good if you know something about the sport you are teaching, that helps in general." (ID_08)

Furthermore, the interviewees (*n* = 7) recommended that trainers of SGPSY should know about mental disorders or at least have a certain understanding of the disorders:

"With that, I mean, also know the disease patterns and be able to say this is within the scope of such-and-such." (ID_10)

*3.2.3.2. Methodological expertise.* Statements regarding the characteristics that might be allocated to metacognitive skills and cognitions as well as acting in different learning and teaching situations (learning & teaching methods) were summarized under methodological expertise [36, 38, 71]. Trainers reported behavior patterns that were summarized as a subcode in relation to transformational leadership as an essential characteristic of successful SGPSY trainers (*n* = 13). As specific aspect of transformational leadership, they referred to a role model behaviour that a trainer adopts and that is having positive effects on participants. The aim of the trainer is to unfold the potential of the participants in the sport setting [51, 72]. The most frequently named (*n* = 9) dimension of transformational leadership behaviour was that the trainer shows *inspirational motivation towards the participants* of the SGPSY, which they described as the trainer being self-motivated and communicating it to the group:

"And I also believe in commitment and enjoyment of the movement. That you can also pass that on to the participants. That you are yourself motivated." (ID_12)

*3.2.3.3. Self expertise.* The subsequent competence that will be discussed is the self expertise which refers to the adequate handling of oneself [36]. Almost half of the trainers (*n* = 7) said they enjoyed sport and being a trainer. Simultaneously, they pointed out the importance of the way this enjoyment was conveyed to the participants.

"Yes, of course she must enjoy doing sports herself. So, I also liked to lead, I also liked to participate myself and was not someone who just stood by and only delegated or led, I think that's very important." (ID_09)

Furthermore, the interviewed trainers mentioned frequently (*n* = 7) that a personal, humanistic view was necessary, according to which IMHD were not primarily perceived as individuals with a disease but as human beings and were accepted as they were:

"I have often had good experiences in which I was totally unprejudiced and simply thought oh here comes a climber, without consciously thinking that it was a person with a mental impairment, which sometimes simply helps. In other words, not the diagnosis-way-of-thinking. It certainly helps in the background, when a situation arises, that you have a background knowledge, but basically also just the openness. Openness and seeing the other

person as a human being and respecting the situation, I believe. And not "Ah, there is now someone who has a burnout and can no longer manage his everyday life, and yes, the poor thing"." (ID_06)

This quote also points out the framework of SGPSY and that they should function as a physical activity that allows IMHD to have positive experiences and lead to positive effects without having a therapeutic framework attached.

*3.2.3.4. Social skills.* Aspects of social skills were named by all of the trainers interviewed (*N* = 15) as a crucial characteristics for a trainer of SGPSY. Social skills refer to the relevant skills for managing social interactions [36, 71]. Personality traits that are necessary to cope with social interactions were mentioned by all interviewees (*N* = 15). Empathy was highlighted most frequently. It also included assessing the IMHD's ability, mental state and sensitivity to the mood within the group:

"Yeah, you maybe need to have a certain sensitivity, to have a little bit of a radar as to where people are standing in a given moment. Because they often have those ups and downs."(ID_02)

Moreover, trainers named flexibility, creativity, tolerance, patience, authenticity, being spontaneous, having a sense of humor, resilience, being calm, and composition as important personality traits for a SGPSY trainers.

In addition to personality traits, handling IMHD was highlighted by several trainers (*n* = 11) as an important social skill. On the one hand, the interviewees explained that anticipative and proper reaction to the participants' behaviour and communication was essential. Mostly it was related to the situation where there was no or very little feedback from IMHD during sports lessons:

"And I think the biggest difference from a "normal" sports group, yes that's always a bit of a bad thing to say, to a group with mentally impaired people is actually that you don't get any feedback. So, a prime example is that you see that they can't really go on anymore, and then you call out to the group "Hey, are you guys still with me?" And I would argue that with any other sports group, at least one shouts "hey, no, I can't anymore" or "yeah sure, I still have much energy left". And here you just do not hear anything. I think that the trainer has to be able to deal with that." (ID_09)

On the other hand, it was frequently mentioned (*n* = 11) that for the trainers, it might be important to handle the challenging behaviours of some IMHD adequately. For example, when participants are limited in their perception or overestimate their skills and abilities:

"And especially for climbing, it's safety-relevant, I believe particularly for climbing it's super important that you give them this looseness, that you give them this space, but that at the same time you also remain tough on the safety-relevant things and perhaps also somewhat reduce their perception. And don't give in on that. Because then they say very quickly. That's exactly where I noticed that some of them can not assess it so well. Because they have never done that before and they think to themselves, exactly what I described before: "I can actually do that and I am not a toddler, I can find my way around." And then they have like the feeling of climbing, again because of their surroundings, because they are all climbing on their own, but also because they have been doing it maybe for years. So, as a trainer, you

just have to always stick closely and consistently to it. Exactly, this balance, I would say, a leader must have." (ID_03)

Further, the interviewees (*n* = 8) mentioned the ability to set boundaries as another aspect of social skills. The trainers described that they considered it important that certain situations did not take them over too much and that they were able to draw clear boundaries and did not take part too much in the private lives of IMHD. In this context, the trainers also explained that the ability to delimitate was of great importance when (some) participants showed, for instance, that they did not appreciate the job of the trainers or even insulted them:

"You also have to be able to handle it when someone says: "The sports lessons are full of shit. You treat me unfairly." You cannot let that get too close to you." (ID_14)

*3.2.3.5. Negative characteristics of a trainer.* In addition to the four competencies presented above, all but one interviewees (*n* = 14) mentioned a characteristic that SGPSY trainer should not have. However, the characteristics were very heterogeneous (e.g., high demand towards oneself; "helper syndrome", etc.), and only a few interviewees mentioned each. "An end in itself" was, however, highlighted as an inappropriate characteristic with regard to the reason to be a trainer by every third person (*n* = 5):

"And otherwise, yes I mean, if it is because of the end in itself, you do not have to do it." (ID_04)

*3.2.3.6. Personal characteristics only rarely named.* Additionally, there were important personal characteristics for a trainer of SGPSY that were named only by a few trainers. Three trainers (*n* = 3) said they considered the ability to create a non-competitive atmosphere within the group (with no performance expectation) as a great asset to a trainer. However, some trainers reported on the contrary that it seemed important for a coach to show some performance expectation of the participants, also with regard to the participants' ability to learn how to deal with failure but with taking into account individual possibilities. Furthermore, attitudes and values–commitment and passion towards the activity–were mentioned three times as necessary characteristics (*n* = 3).

**3.2.4. Reasons and motivation to be a trainer of SGPSY.** The reasons and motivations of the trainers are summarized in Table 5 and described below. These can be divided into two groups of reasons for volunteering which emerged in previous research in volunteering with individuals with mental health disorders or volunteering within the framework of sports

**Table 5. Frequencies of reasons and motivations to be a trainers of SGPSY.**

| Reasons and motivations | Frequency | Subcategory | Frequency |
|---|---|---|---|
| Target group | 14 (93%) | | |
| | | *Dealing with & understanding of IMHD* | 9 (60%) |
| | | *Maintaining, promote movement of IMHD* | 9 (60%) |
| | | *Enable IMHD to succeed* | 5 (33%) |
| Doing something meaningful & valuable | 7 (47%) | | |
| Pleasure & fun | 11 (73%) | | |
| Studies of Psychology | 5 (%) | | |

*Note*. Full sample *N* = 15; Reasons and motivations mentioned less than five times are not included

groups: On the one hand, literature shows that persons do volunteer to do something for others and, on the other hand, to do something for oneself [46, 73].

*3.2.4.1. Doing something for others.* Almost all trainers (*n* = 14) stated that they were involved as a trainer because of the target group, the IMHD. Several trainers said they were trainers to promote the physical activities of IMHD. They were therefore committed to their welfare. The trainers indicated that they were convinced that physical activity was good for IMHD and wanted to contribute to the target group's increased participation in physical activity.

> "They are people who are on the very path to integration, and I think that is why this is a good offer in order to support them. And I think sport has a positive influence, which is also quite well proven, and therefore it is a good service to commit oneself to it." (ID_08)

Furthermore, one-third of the trainers (*n* = 5) said that they pursued the activity because they wanted to enable IMHD to achieve success through participation in the voluntary sports groups for IMHD (SGPSY). From the interviewees' point of view, trainers were committed to ensuring that the IMHD participated in the SGPSY, that they achieved something for themselves, were able to build a daily structure and, create a place where these successes could be celebrated.

Moreover, almost half of the trainers (*n* = 7) described that they were in the position of a trainer since they wanted to do something meaningful and valuable. In other words, they wanted to do something for others and intended to give something back to society in this way, thus referring to social responsibility towards society.

> "So, because I'm sure the people who are not so on the sunny side of life, because they are close to my heart and because I think sport is a cool way to give them normality or an anchor in their already quite difficult lives. That they like, also can get a little bit of a support group or that they can have the exchange with us, who are kind of doing well and who also have a lot of resources or patience to help them like that, also outside of the sport offer. I also see that as a bit of everyone's responsibility. It doesn't necessarily have to be for mentally impaired people per se, but the world is certainly better if you help some group of people who are in need of it, and that's just now through my studies, where I'm very interested in particular mentally distressed people, I've chosen like this group to invest my time, my free time also a little bit. To help the people who particularly interest me. For others, it might be children or the poor or whatever." (ID_14)

*3.2.4.2. Reasons to do something for yourself.* According to the interviewees, the further motivation and reasons for an activity as a trainer of SGPSY were the related experience and benefits for themselves. The trainer (*n* = 9) emphasized that through their role, they gained a better comprehension of IMHDs by understanding their perspective and also gained experience in dealing with this target group. This subsequently led to their acquisition of new skills.

> "And I think in the longer term it gives me a feeling of already having some kind of acquaintance with people with mental disabilities. That means it also helps me for my studies and gives me a better understanding of how to deal with people with mental disabilities. But I also think that somewhere I find it quite cool to have a practical insight in addition to the theoretical aspect of my studies." (ID_12)

Some respondents (*n* = 5) connected this kind of reason to their ongoing education or job (e.g., psychology students). They indicated that they were doing this to apply what they have

learned in their studies. Some also mentioned that they had started this position because it was possible to get (internship) credits for their studies. However, all of these respondents highlighted that they kept training the group after the necessary duration of the educational internship.

Furthermore, trainers (*n* = 11) stated that they got involved because they themselves enjoyed the activity, liked doing it and passing it on:

"Yes and it's just fun, climbing is just my hobby, my passion and my profession." (ID_06)

"On the other hand, because I also like being in a leadership position and passing on to people something I enjoy." (ID_09)

"And thirdly, because I myself have a lot of fun with movement and think it's cool to pass that on to others." (ID_12)

*3.2.4.3. Reasons and motivations only named rarely*. The financial compensation was mentioned twice (*n* = 2) as a possible reason for the work as a coach in SGPSY.

**3.2.5. Effects of the trainer experience.** The research question was focused on the effects the activities have on the trainer. The aim was to find out if there were any positive/negative experiences for the trainer and what they felt after working in SGPSY. Table 6 shows the summarized results. The description is presented below.

*3.2.5.1. Satisfaction and gratification*. Except for two respondents (*n* = 13), everyone mentioned a feeling of deep satisfaction after guiding the training. The pleasure of being in physical action and being able to pass it on was also mentioned by the participants.

"A volunteer said "you know, it takes a bit of effort to get up on a Saturday morning and come to the climbing gym to help out and not climb myself, but I go home with a smile every time". And that's how I feel. I like to make the offers, but it's still a Saturday morning where you don't go ski touring or whatever, or climbing for yourself. But I go home every time with a happy feeling, in the sense of "we had fun together, we could be proud of each other, we had a good time together, we had great experiences and see that many people go home strengthened". That is already what gives a lot of satisfaction." (ID_05)

*3.2.5.2. Witness of the participants success*. Moreover, almost half of the trainers (*n* = 7) mentioned that they witnessed their participants' successes through their support (as trainers). This observation gave them a positive feeling as well. They stated that seeing the successive physical improvements of the IMHD as well as the increase in their well-being had a positive impact on them:

"Just things that develop and being able to witness that, that's really really nice. I can perhaps give you an example. There are people who hardly dare to say anything and then

**Table 6. Frequencies of the effects of the trainers experience.**

| Effects | Frequency |
|---|---|
| Satisfaction and gratification | 13 (87%) |
| Personal development | 11 (73%) |
| Witnessing participants' success | 7 (47%) |

*Note*. Full sample *N* = 15; Effects mentioned less than five times are not included

maybe have to stand in a circle and then maybe you play a little bit with the ball and then do exercises where everyone who takes the ball calls out "yes" and then when someone suddenly calls out "yes" or says "yes" who otherwise never makes a sound and then to see what happens to them, that's just super cool and simply nice" (ID_04).

*3.2.5.3. Personal development.* In addition, most trainers (*n* = 11) mentioned their personal development as another impact. They explained that their role made them more patient, calm, open, experienced, self-confident, sensitive and understanding. In addition, they were able to get to know themselves better through their work and acquire new skills, also in understanding other people:

"And also a great enrichment, it trains you extremely as a person. So you become more sensitive, maybe you also become a bit more open, more understanding. Yes, and I still find climbing super cool. I love these movements and the fact that I work with these people also trains me as a leader. So, I can also benefit extremely from them, because I still have situations where I'm just not equipped for it and then just try to get the best out of it and sometimes it works and sometimes not. And when you don't succeed, you just reflect on it and ask yourself what you can do better next time." (ID_06)

*3.2.5.4. Effects on trainers only mentioned rarely.* Besides the effects mentioned only once or twice (e.g., being in the flow), several other effects were also discussed by the minority of participants. Some trainers (*n* = 4) said that the appreciation of the IMHD for the work of the trainer gave them a good feeling and confirmed that they were doing something good. Furthermore, other interviewees (*n* = 3) mentioned that through their work in SGPSY (and contact with IMHD), they were able to put their everyday problems into another perspective.

At the same time, some interviewees (*n* = 3) also described that the function had opposite effects on them, making them sometimes pensive or even worried about their participants or distressed after their work as a trainer. That mostly happened when they started to think about the difficulties their participants faced in their lives (e.g., emotional instability, mental stress).

**3.2.6. Other topics.** In the interviews, topics were raised or described that cannot be assigned to any research questions presented here. However, these topics can be of importance for the further development of sports offers for IMHD and the education of those trainers:

The majority of interviewees (*n* = 11) criticized the view on and the perception of IMHD in society. They said that IMHDs were often stigmatized or viewed as immature and that mental disorders were still considered taboo in society.

Five trainers (*n* = 5) were of the opinion that there are stil not enough voluntary sports programs for IMHD in CH. However, their expansion in the passed few years was also emphasized:

"It was just a while ago, about 15 years ago, [. . .] it was taboo until then, so people didn't address mental impairments. It was ignored, it didn't exist, it wasn't part of a target group. And that has changed massively. And yes, I've known Nikolai for a long time, and he really embraced this topic and is pushing it. And I think that's great, of course, and I think that's why there are more and more sports groups. It is still a very small proportion if you compare it with other sports groups. There is this overview, which I have looked into. And yes, there are of course more other sports groups, there is ProCap, PluSport, Insieme and all the offers in the institutions themselves. It's still a disproportion, but it's coming about and I think that's great. And for that, of course, you need people who lead these sports programs." (ID_08)

Four trainers ($n$ = 4) commented on the issue of education in organizations providing sports for IMHD. They gave suggestions for improvement that might be crucial for the positive and sustainable development of the SGPSY:

"And maybe what I could also say as a suggestion for improvement: I sometimes have the feeling that the training courses are very strongly focused on physical and mental impairments. And I think it would be even cooler if they could focus more specifically on dealing with people with mental disabilities. Or perhaps, those who do this could be brought together in a group, so that they could also exchange ideas." (ID_12)

## 4. Discussion

Previous research has established strong evidence for the importance of physical activity for individuals with mental health disorders. So, in psychiatric hospitals in Switzerland, physical activity is an important part of the treatment program [18]. However, outside psychiatric hospitals there are only a few offers for IMHD to do sports if they are unable or not prepared to participate in regular physical activities offered by the classical mass sports system [22, 24]. So far, only one organization offers specialized voluntary sports groups for individuals with mental health disorders in CH. Until now, there was almost none information about the functionality and needs (of trainers and participants) within those groups. Therefore, the trainers of these voluntary sports groups were interviewed in the present study to explore the needs of the trainers (of SGPSY), the required personal characteristics of these trainers, their reasons and motivations, and the effects they perceive from being a trainer on they own from a trainers point of view. The results of the present exploration might be crucial to ensure and increase the quality of the sports groups and programs and for the future development of the sports offers for IMHD. Moreover, the results should be implemented to update the physical activity programs to the needs of the users in order to increase the utilization of the sports by IMHD.

Regarding the participants of the SGPSY, those aspects made it difficult for IMHD to participate. Looking at those challenges and barriers for participation, the results presented confirm that the mental state as well as motivation and lack of drive were highlighted most frequently from trainers and thus confirmed results of studies in ambulatory as well as in hospital settings [12, 20, 50]. Hence, trainers described the importance to show inspirational motivation to face this issue–so that the trainer him or herself showed motivation and could transfer it to the participants, as it is shown in the behaviour of transformational leadership, thus confirming literature [49, 51]. Moreover, the study showed that the help of the environment impacted the participation of IMHD for the SGPSY–on the one hand persons in their near environment, and on the other hand, the institutions they lived in. So, it seems crucial to include the involved parties.

Further, constancy was highlighted as an enabler to participation and a needed characteristic of trainers. Interviewees explained that a constant group composition, constant trainers and a constant procedure of the sport groups was likely to increase the probability of IMHD participating in the activity. This supports the findings that constancy is an important factor for helping IMHD to participate in sport activities [25, 27].

Importantly, the trainers stated that they have to be able to face challenges that are related to the composition of the group. As this could be a hindering factor as well as an enabler for participation, it seems important, that in the education of the trainer the handling and the promotion of such issues is included, which was also named as needed charactersitics of a trainer. Within this, also personality characteristics of the social competences, such as empathy, are important. This was not named as such in the literature and brings new insights.

Furthermore, the study revealed reasons and motivations why trainer pursue the activity as well as the effect it had on them. This information is valuable for further recruitement of trainers of this groups, as it is expected that the number of this kind of sport groups is going to constantly grow (Kiselev & Loosli, 2019). On the one hand trainers reported that they were in their function to do something for themselves, for their wellbeing and to develop themselves. In relation to that, trainers reported that after giving their SGPSY they felt well and satisfied, which supports literature regarding volunteering activities [52, 54, 74]. They further explained that they appreciated to see the success IMHD achieved (e.g., enhance their physical fitness; being able to participate), and the positive effects it had on the trainer themselves. However, trainers also named negative effects they sometimes felt as a result of the sport lesson; for instance taking part in the lives of their participants. So, within this, the capacity to delimitate oneself from the participants, which can be put into combination with a healthy empathy, is very important. That is, only with a good delimitation will the trainer be able to be a good trainer and support the participants. As his/her function is clearly different from the therapist-function of the IMHD. On the other hand trainers explained that they were in their function to do something for others: in this case for individuals with mental health disorder and also due to social responsibility towards the society. This often also offered positive feelings after the sport activity, trainers reported.

The results of the study are important for the recruitment of trainers in the sense that important characteristics, as some personality characteristics (social competence) and the reasons why they are in their function, can help to search suitable persons as well as within a certain population (e.g. persons which are interested in the target group).

## 4.1. Limitations

The applied qualitative approach was applied to obtain new insights in the topic of trainers of voluntary sport groups for individuals with mental health disorder (SGPSY) as it is the first one, known at the present moment. This approach brings new insights into a topic only few researched, which is presented here. However, this approach also has some limitations. Due to the small sample and the subjective data analysis, statistical generalisation and thus generalisation is not or only possible within limits [57, 64, 75]. However, a detailed procedure was applied and followed to face this issue. Furthermore, the explorative design is characterized by its subjective approach, so the analyses were influenced by the researchers, which includes a potential bias, as postulated by researchers [57, 75]. This means that the researchers themselves may, for instance, had expectations of the outcomes to which they are more open and more likely to find [57, 75]. However, this was addressed throughout the whole process of the study by following recommendations from different authors, as the researchers meticulously followed the previously presented guideline [57, 75].

Moreover, the trainers who were interviewed were the first trainers of SGPSY for IMHD and it is unclear whether the sample will be representative of trainers in a few years' time, as it is expected that the number of SGPSY multiplies in the following years [32, 56]. So, within this, it is crucial to do further research to confirm the results.

## 5. Conclusion

The present study revealed for the first time insights in the context of voluntary sport groups for inidividuals with mental health disorder in Switzerland. The further promotion of physical activity in IMHD is important and was supported from different authors [2, 18]. The SGPSY want to promote the maintaining physical activity. The findings highlight that it is important that trainers get a good education in order to be prepared for the difficulties IMHD face

regarding participation (e.g. motivation, listlessness, etc.) and to promote enablers (involvement of institutions and persons in the close environment). Furthermore, the knowledge of the reasons as well as effects of the function as trainers can help for the recruitement of future trainers. More research is needed to confirm the presented results and especially also from the point of view of participants of those groups.

## 6. Current recommendations

Based on the findings and discussion above, several recommendations can be made to the Swiss disabled sports systems (SDSS) that can enhance voluntary sports groups for individuals with mental health disorders, foster participation, promote well-being, and achieve positive outcomes for both trainers and participants.

First, SDSS should continuously provide comprehensive education and training programs for trainers of SGPSY. These programs should focus not only on sports-related knowledge and skills but also on understanding mental health disorders, effective communication, empathy, and strategies for promoting participation and well-being. Second, within this education, the importance of transformational leadership behaviors among trainers should be emphasized. Trainers should serve as role models, inspiring and motivating participants through their own enthusiasm for physical activity. This can help address motivational barriers and enhance participants' self-efficacy. Third, at the same time, trainers of SGPSY should be trained in setting boundaries and maintaining appropriate delimitation between their role as trainers and the therapeutic aspects of participants' lives. This helps ensure a professional and supportive environment while avoiding potential over-involvement or blurred boundaries. Fourth, the recruitment of further SGPSY trainers should focus on personal characteristics such as empathy, patience, flexibility, tolerance, resilience, and a humanistic view of IMHD. These characteristics contribute to creating a supportive and inclusive environment that fosters participants' well-being and positive experiences. Fifth, trainers of SGPSY should strive to promote consistency of group and trainers composition and procedures within the sports groups. This consistency helps establish a sense of safety and familiarity for participants, increasing their likelihood of continued participation. Sixth, trainers and disabled sports clubs should encourage collaboration and involvement of institutions and individuals in the participants' close environment, such as family members, friends, and mental health professionals. Their support and understanding can play a crucial role in facilitating participation and providing a positive social network for participants. Seventh, the availability and diversity of SGPSY should be increased. This expansion should include collaborations between sports organizations, mental health institutions, and community resources to provide a broader range of options and opportunities for participation. This might be a low-intence and cost-effective solution to improve the vast number of persons suffering from mental health disorders. Finally, further research to explore the experiences, needs, and perspectives of individuals with mental health disorders who participate in voluntary sports groups should provide valuable insights for program development, tailoring interventions to their specific needs and enhancing the overall effectiveness of these programs.

## Author Contributions

**Conceptualization:** Florence Epiney, Frank Wieber, Daniela Loosli, Nikolai Kiselev.

**Data curation:** Florence Epiney, Daniela Loosli, Nikolai Kiselev.

**Formal analysis:** Florence Epiney, Daniela Loosli, Nikolai Kiselev.

**Funding acquisition:** Frank Wieber, Daniela Loosli, Nikolai Kiselev.

**Investigation:** Florence Epiney, Daniela Loosli, Nikolai Kiselev.

**Methodology:** Florence Epiney, Frank Wieber, Hansjörg Znoj, Nikolai Kiselev.

**Project administration:** Daniela Loosli, Hansjörg Znoj, Nikolai Kiselev.

**Resources:** Florence Epiney, Daniela Loosli.

**Software:** Florence Epiney, Nikolai Kiselev.

**Supervision:** Frank Wieber, Hansjörg Znoj, Nikolai Kiselev.

**Validation:** Florence Epiney, Frank Wieber, Daniela Loosli, Nikolai Kiselev.

**Visualization:** Florence Epiney, Nikolai Kiselev.

**Writing – original draft:** Florence Epiney, Frank Wieber, Daniela Loosli, Hansjörg Znoj, Nikolai Kiselev.

**Writing – review & editing:** Florence Epiney, Frank Wieber, Daniela Loosli, Hansjörg Znoj.

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
