## [Decision Letter · Decision Letter 0]

19 Jun 2023

PONE-D-23-01364Voluntary sports programs for individuals with mental health disorders: the trainer's viewPLOS ONE

Dear Dr. Nikolai Kiselev,

Thank you for submitting your manuscript to PLOS ONE. After careful consideration, we feel that it has merit but does not fully meet PLOS ONE’s publication criteria as it currently stands. Therefore, we invite you to submit a revised version of the manuscript that addresses the points raised during the review process.

We look forward to receiving your revised manuscript.

Kind regards,

Rogis Baker, Ph.D

Academic Editor

PLOS ONE

Journal Requirements:

Reviewers' comments:

Reviewer's Responses to Questions

**Comments to the Author**

1. Is the manuscript technically sound, and do the data support the conclusions?

Reviewer #1: Yes

Reviewer #2: Yes

2. Has the statistical analysis been performed appropriately and rigorously? 

Reviewer #1: Yes

Reviewer #2: Yes

3. Have the authors made all data underlying the findings in their manuscript fully available?

Reviewer #1: Yes

Reviewer #2: Yes

4. Is the manuscript presented in an intelligible fashion and written in standard English?

Reviewer #1: Yes

Reviewer #2: Yes

5. Review Comments to the Author

Reviewer #1: Interesting article.

Suggestions and comment:

1. In methodology - suggest to describe about the interviewer in more detail. E.g. how many interviewer, which year are they, are they homogenous group of students and are they part of the study team

2. Recruitment- Please explain or reword "corona time"

3. Do you apply for ethic clearance?

4. I don't think you need to mention regarding the gift etc in the text

5. Please reword/ rephrase "Msc in psychology" not everyone familiar with that

6. Table 1: spelling mistake

Reviewer #2: The article has good data , although the authors have said that they done need ethical approval for such publications i would recommend that ethical approval be taken ,

The authors are advised to add some recommendations based on the finding.

6. PLOS authors have the option to publish the peer review history of their article (what does this mean?). If published, this will include your full peer review and any attached files.

Reviewer #1: No

Reviewer #2: **Yes: **Dr Mohammed Feroz Ali

---

## [Author Response · Author response to Decision Letter 0]

22 Jul 2023

Dear Reviewers,

Thank you for your time you spent reviewing our manuscript as well as your recommendations. 

You can find our answers attached. All of your recommendation have been implemented in the manuscript.

All the best

Authors

---

## [Editor Report · Decision Letter 1]

8 Aug 2023

Voluntary sports programs for individuals with mental health disorders: the trainer's view

PONE-D-23-01364R1

Dear Dr. Nikolai Kiselev

We’re pleased to inform you that your manuscript has been judged scientifically suitable for publication and will be formally accepted for publication once it meets all outstanding technical requirements.

Kind regards,

Rogis Baker, Ph.D

Academic Editor

PLOS ONE
---

## [Editor Report · Acceptance letter]

11 Aug 2023

PONE-D-23-01364R1 

Voluntary sports programs for individuals with mental health disorders: the trainer's view 

Dear Dr. Kiselev:

I'm pleased to inform you that your manuscript has been deemed suitable for publication in PLOS ONE. Congratulations! Your manuscript is now with our production department. 

Kind regards, 

on behalf of

Dr. Rogis Baker 

Academic Editor

PLOS ONE